# Sensitivity Analysis of Italian *Lolium* spp. to Glyphosate in Agricultural Environments

**DOI:** 10.3390/plants9020165

**Published:** 2020-01-30

**Authors:** Silvia Panozzo, Alberto Collavo, Maurizio Sattin

**Affiliations:** 1Institute for Sustainable Plant Protection (IPSP) – CNR, Viale dell’Università 16, 35020 Legnaro (PD), Italy; silvia.panozzo@ipsp.cnr.it (S.P.); alberto.collavo@bayer.com (A.C.); 2Bayer AG, Industriepark Hoechst, H872, D-65926 Frankfurt am Main, Germany

**Keywords:** sensitivity line, ryegrass, herbicide dose, herbicide resistance, dose-response

## Abstract

Empirical observations generally indicate a shifting and decreased *Lolium* spp. susceptibility to glyphosate in Italy. This is likely due to the long history of glyphosate use and to the sub-lethal doses commonly used. There is, therefore, a need to determine the variability of response of *Lolium* spp. to glyphosate and identify the optimum field dose. To perform a sensitivity analysis on *Lolium* spp. populations in an agriculture area, collection sites were mainly chosen where glyphosate had not been applied intensely. Known glyphosate-resistant or in-shifting populations were included. Two outdoor dose-response pot experiments, including eleven doses of glyphosate, were conducted. The dose to control at least 93%–95% of susceptible *Lolium* spp. was around 450 g a.e. ha^−1^. However, to preserve its efficacy in the long term, it would be desirable not to have survivors, and this was reached at a glyphosate dose of 560 ± 88 g a.e. ha^−1^. Taking into account the variability of response among populations, it was established that the optimal dose of glyphosate to control *Lolium* spp. in Italy up to the stage BBCH 21 has to be at least 700 g a.e. ha^−1^. As a consequence, it is recommended to increase the label recommended field rate for *Lolium* spp. control in Italy to a minimum of 720 g a.e. ha^−1^.

## 1. Introduction

*Lolium rigidum* Gaud. (rigid ryegrass, LOLRI) and *Lolium multiflorum* Lam. (Italian ryegrass, LOLMU) are two self-incompatible species that have a global distribution [1]. According to reported cases [2], they are among the species most prone to evolve herbicide resistance. *L. rigidum* is one of the most troublesome weeds in grain crops as well as orchards, olive groves and vineyards, where it is also managed as a cover crop [3]. To date, *L. rigidum* populations resistant to 13 different herbicide Sites of Action (SoA) have been reported [4]. *L. multiflorum* occurs in several temperate countries and populations resistant to nine different herbicide SoA have been reported [4]. Both species originated from the Mediterranean, have a C_3_ photosynthetic pathway and produce dense infestations [5]. The two species are often mixed in the field and not always easily identifiable, and in those cases, the population is defined as LOLSS (*Lolium* species).

The withdrawal of many herbicides from the EU market due to the strict regulation, the lack of herbicides with new SoA [6] and the propensity of *Lolium* spp. to evolve resistance to the most commonly used post-emergence herbicides have increased the importance of glyphosate for the management of these species in agricultural and in non-agricultural areas [1]. Glyphosate is the most successful herbicide in history [7,8], and its use is higher than any other herbicide SoA [6]. According to the HRAC (International Herbicide Resistance Action Committee) classification [9], which is based on the herbicides’ SoA, it belongs to group G and exerts its action by inhibiting 5-enolpyruvylshikimate-3-phosphate (EPSP) synthase in plants, fungi and microorganisms, the only life forms that possess the shikimate pathway. Therefore, it has no measurable mammalian toxicity at the concentrations used [10]. It acts as a competitive inhibitor to the phosphoenol-pyruvate (PEP) binding site and a non-competitive inhibitor for the shikimate-3-phosphate (S3P) site, thus preventing the formation of EPSP [11].

Glyphosate is a non-selective, systemic, post-emergence herbicide that controls many dicotyledonous and monocotyledonous weeds [12]. It is neither active nor residual in the soil and, therefore, selection pressure for resistance is only exerted on emerged seedlings [13]. Glyphosate-resistant weeds were not found during the first 22 years of glyphosate use, whereas in the last 23 years (1996–2019), glyphosate resistance was documented in 45 weed species in 29 countries [4].

Glyphosate is also commonly used on a frequent basis between tree rows (i.e., olives, hazelnuts and vineyards) and for roadside weed control [14]. In these situations, glyphosate can be used for many years and applied several times per year. The continuous use of glyphosate in perennial crops, such as orchards, has imposed intense selection pressure for resistance evolution and has led to shifts in weed floras as well as towards glyphosate-resistant individuals [15,16]. In particular, in Europe, glyphosate resistance has evolved most often in two genetically diverse, but at the same time, resistance-prone genera, *Conyza* [17] and *Lolium* [18].

In Italy, the doses of glyphosate commonly used have become sub-lethal for *Lolium* spp. [19]. In the past, the rate of 360 g a.e. (acid equivalent) ha^−1^ gave satisfactory control, but it is likely that a few plants survived each treatment. Exposure to recurrent selection at sub-lethal glyphosate doses can result in a shift towards resistance within a few generations. It was clearly demonstrated by Busi and Powles [20] that in allogamous species, such as *Lolium*, minor resistance gene trait(s) may be additively enriched through cross-pollination among surviving plants. In Italy, empirical observations indicated a general decrease in susceptibility of *Lolium* spp. to glyphosate (i.e., relatively poor control at 360 g a.e. ha^−1^) and the first resistance cases were reported in 2008 [19]. At the moment, 13 municipalities in five Italian regions and five different cropping systems (including orchards, olive groves, vineyards, wheat and no-tillage agriculture) are affected by glyphosate resistance [21].

As part of the herbicide resistance risk analysis and management, the availability of a robust baseline sensitivity for key-target species is critical to discriminate between susceptible (S) and resistant (R) populations and to identify early shifts in susceptibility. From a practical point of view, a population is ascribed as resistant (R) to a herbicide when more than 20% of treated plants survived the recommended herbicide field dose [22]. The identification of a first shift in susceptibility is particularly valuable when resistance evolution is rather slow, as in glyphosate resistance. Only a few herbicide sensitivity analyses are available in the literature [23,24,25,26]. The European and Mediterranean Plant Protection Organization defines the baseline as the mean of natural variability of a target species’ sensitivity before the commercial introduction of an active ingredient and can be taken as a point of reference to be used in decision-making processes. Instead, glyphosate has been on the market for many years, and its selection pressure has been active for a long time. In such a case, the baseline term/approach is not correct, and a sensitivity analysis should instead be performed. The aim of a sensitivity analysis is to determine the average efficacy of an old herbicide on weed populations that may have been treated before with the same compound [27,28]. In other words, this is part of the monitoring procedure of herbicide efficacy. To our knowledge, only one paper in the literature has dealt with the glyphosate baseline sensitivity for *L. rigidum* in Spain using a quick Petri dishes test [29].

The establishment of a good sensitivity baseline should make it easier to identify any case of evolved herbicide resistance [30] and would have an added value if an effective monitoring program is initiated [29].

The aims of this research were (1) to determine the variability in glyphosate response of *Lolium* spp. populations collected from Italian agricultural environments and (2) to determine the glyphosate dose that is actually effective on the *Lolium* spp. populations in field conditions in order to preserve its efficacy in the long term.

## 2. Results

### 2.1. Dose-Response Experiments

Two outdoor pot dose-response experiments were performed during spring (March-May) and autumn (September-November) to test the effects of increasing glyphosate dose on plant survival and fresh weight for several *Lolium* spp. populations collected from Italian agricultural environments. The effective doses—EDs—and growth rates—GRs—causing 50% and 90% reduction in plant survival and fresh weight (ED_50_-ED_90_ and GR_50_-GR_90_), respectively, were calculated using a regression analysis (see Section 4.3). 

#### 2.1.1. Spring Dose-Response Experiment

Twenty populations were included in the spring experiment (Table 1a). A variance test (F-test) was performed to compare the dose-response curves obtained for the different populations in the experiment. The lack-of-fit F-test on both plant survival and fresh weight indicated that it was not possible to simplify the glyphosate regressions to a model with a common slope for all populations: the slope tended to decrease when EDs (and GR_S_) increased. The data of each population were, therefore, regressed as individual curves and treated separately.

Among the populations included in the spring experiment, eight were known resistant/shifting populations. A Box and Whisker analysis using the median and 25–75 percentiles was used to statistically exclude outliers. The analysis was performed including all populations, then repeated excluding the outliers until no further outliers were identified (Figure 1). The first analysis revealed three extreme value populations (403, 392 and 401) (Figure 1A), all previously confirmed as resistant (Table 1). The analysis was repeated excluding those populations and limiting the ED_50_ range to between 155 and 900 g a.e. ha^−1^. In this second step, two other populations (343 and 384L) were found to be outliers (Figure 1B). The third analysis considered an ED_50_ range of 155–560 g a.e. ha^−1^ and highlighted three other outlier populations (384, 259 and 328) (Figure 1C), one included in the experiment as R check (384) and the two shifting populations (Table 1). The fourth analysis with 12 populations and an ED_50_ range of 155–260 g a.e. ha^−1^ did not reveal any outlier population (Figure 1D). The Box and Whisker analysis, also repeated for the ED_90_ and GRs values (data not shown), confirmed that all eight populations included in the experiment as resistant or partially resistant to glyphosate had a reduced sensitivity or resistance to this herbicide.

Considering the data of the other twelve populations resulted as being sensitive to glyphosate, it was highlighted that glyphosate ED_50_ ranged from 155 ± 5.9 to 260 ± 6.7 with a mean value of 206 g a.e. ha^−1^ and ED_90_ from 243 ± 20.8 to 506 ± 79.1 with a mean value of 342 g a.e. ha^−1^. Concerning fresh weight, GR_50_ varied from 31 ± 8.8 to 98 ± 14.7 with a mean value of 64 g a.e. ha^−1^, GR_90_ from 144 ± 15.7 to 272 ± 26.3 with a mean value of 198 g a.e. ha^−1^.

#### 2.1.2. Autumn Dose-Response Experiment

Twelve susceptible populations selected in the spring experiment were also tested in the autumn experiment, together with another eighteen populations reported in Table 1b. In this second experiment, the Box and Whisker analysis did not highlight any outliers among populations. 

As for the spring experiment, the lack-of-fit F-test on both plant survival and fresh weight in the autumn experiment indicated that it is not possible to simplify the regressions to a model with a common slope for all populations, so a single-curve analysis was preferred. 

Furthermore, an ad hoc lack-of-fit F-test performed on the data of each population included in both experiments comparing the dose-response curves obtained in the two experiments showed that most of the curves were significantly different at *p* < 0.05, so the two experiments cannot be merged (data not shown). Populations data could not be pooled considering both plant survival and fresh weight, so it was decided to consider the two experiments separately. 

ED_50_ based on the autumn dose-response experiment ranged from 108 ± 10.1 to 282 ± 7.1 with a mean value of 186 g a.e. ha^−1^ (Figure 2a) and ED_90_ from 189 ± 16.7 to 561 ± 87.7 g a.e. ha^−1^ with a mean value of 317 g a.e. ha^−1^ (Figure 2b); concerning fresh weight, GR_50_ varied from 37 ± 4.8 to 148 ± 8.4 with a mean value of 78 g a.e. ha^−1^, GR_90_ from 136 ± 10.3 to 295 ± 37.6 g a.e. ha^−1^ with a mean value of 199 g a.e. ha^−1^ (data not shown). 

Cluster analyses were used to determine whether correlations were present between the calculated parameters (i.e., EDs and GRs) and the collection sites or geographical origin as well as species of the different populations. A cluster analysis based on the ED_50_ highlighted two clusters (Figure 2a), whereas the data were divided into four clusters when ED_90_ were considered (Figure 2b). In both cases, no correspondence was detected among these divisions and geographical origin of the populations, *Lolium* species or cropping system/collection site. Similar results were obtained considering GRs; therefore, data are not reported.

### 2.2. Sensitivity Line Calculation 

Based on the results obtained through the dose-response experiments, the range of glyphosate susceptibility of *L. multiflorum* and *L. rigidum* sampled in Italian agricultural environments was established, a sensitivity line was calculated and the dose of glyphosate to fully control *Lolium* spp. in agronomic conditions was proposed.

The paired t-test proved that no significant differences were found between the mean EDs and mean GRs of the two experiments (considering only the common populations), so for calculation of the threshold value of the sensitivity analysis, only the data of the autumn experiment, having a higher number of populations, were considered. 

Based on the mean value of ED_90_ as well as the variability across and within populations (Figure 2b), it was established that an agronomically suitable dose (i.e., at least 93%–95% of control) to adequately control susceptible *Lolium* species was around 450 g a.e. ha^−1^ of glyphosate. Therefore, the current dose indicated in Italy (480 g a.e. ha^−1^) is enough to adequately control susceptible plants, confirming that the old dose (360 g a.e. ha^−1^) was sub-lethal for many populations. As an anti-resistance measure, it is important to keep efficacy at or near 100% to avoid, or at least slow down, the selection and eventually the evolution of glyphosate resistance under tough climatic conditions or weed growth stages that can affect glyphosate efficacy level.

Figure 3 demonstrates how the sensitivity analyses data can be used to identify potentially resistant populations. The range of ED_90_ for the autumn experiment was 189-561 g a.e. ha^−1^, with a mean sensitivity line of 317 g a.e. ha^−1^. A population can, therefore, be considered as shifting (or partially resistant) if the difference between the threshold value and population is greater than 2x (634 g a.e. ha^−1^) [26] (e.g., populations 259 and 328 included in the spring experiment). A population can be considered as resistant if the difference between threshold value and population is greater than 3x (951 g a.e. ha^−1^) (e.g., populations 384, 384L, 343, 392, 403 and 401 included in the spring experiment, plus populations 332 and 336 [31]) (Figure 3).

Two parameters that illustrate the variability of the response of populations to glyphosate were calculated (see Section 4.4). ED_50/90_ variations do not fully explain the overall variability, and slope also has to be taken into account (Figure 4). When the ratio is close to one the slope tends to be vertical, i.e., small variations of glyphosate dose around EDs cause large variations in weed control. However, we did not observe any relation between collection site and slope. The Sensitivity Index (S.I.) proved to be three, demonstrating that there is a three-fold difference in sensitivity to glyphosate between *Lolium* spp. populations harvested across Italian agricultural environments.

## 3. Discussion

Glyphosate is an efficient herbicide, and the evolution of resistant weeds is a big hindrance to efficient control in many circumstances [7]. Given that no herbicides with truly new molecular target sites have been marketed in the past 30 years and that there is no silver bullet chemistry ready to enter the marketplace [32], glyphosate efficacy should be preserved in the long term, especially in those cropping systems where there is a shortage of post-emergence herbicides (i.e., targeting grasses) or as a tool in weed resistance management. For these reasons, efficacy of herbicide treatments should be kept at or near 100% to avoid or at least slow down the selection and eventually the evolution of glyphosate resistance.

The two dose-response experiments performed to calculate the sensitivity line of glyphosate in Italian agricultural environments could not be pooled together for several reasons. First of all, in the second experiment, a larger number of populations coming from different parts of Italy and different agricultural systems were included in order to give a higher impact to the study. Secondly, it is very rare that two experiments conducted in outdoor conditions can be considered together because there are too many uncontrollable variables (e.g., variation in temperature, rainfall). In particular, this was expected using glyphosate because its performance is known to vary seasonally [18,33].

On the basis of ED_50_, GR_50_, ED_90_, GR_90_ and slope, it was not possible to discriminate *L. multiflorum, L. rigidum* or intermediates, and neither a difference related to geographical areas nor collection site was found. Most probably, if a correlation is present among these values and the variables considered, a more specific study with a larger number of populations needs to be assessed. An example was reported for 80 accessions of *Echinochloa* spp. where *E. crus-galli* was found to be more sensitive than other *Echinochloa* species when sprayed with azimsulfuron or cyhalofop-butyl [23].

On average, GR_50_ and GR_90_ (78 and 199 g a.e. ha^−1^, respectively) were lower than ED_50_ and ED_90_ (186 and 317 g a.e. ha^−1^, respectively) indicating that a significant proportion of surviving plants had a low fitness and likely a low competitivity with the crop. However, it cannot be excluded that they could produce some seeds [34]. Therefore, in order to not underestimate this aspect, the ED values were used to calculate the sensitivity line.

The Box and Whisker analysis indicated that the selection of populations was adequate for the purpose of the study; in fact, only the populations included in the spring experiment as resistant or partially resistant checks were excluded through the analyses (Figure 2), whereas no outliers were found in the autumn experiment. 

Data variability increased with EDs, which may indicate that less susceptible populations are also less homogeneous in terms of glyphosate susceptibility. The proposed field dose also considers this variability and was calculated excluding the eight outlier populations identified through the Box and Whisker analysis in the spring experiment. The eight outliers correspond to the resistant and partially resistant (i.e., in-shifting) populations included for comparison (Table 1a) and for which the resistance mechanisms have been described elsewhere (Table 1a [19,31]). In this study, it was demonstrated that the optimal glyphosate dose to control *Lolium* spp. in Italy at the growth stage of first shoot visible (i.e., using the Extended BBCH scale at growth stage 21 [35]) should be 700 g a.e. ha^−1^ of glyphosate or higher. Indeed, in our experimental conditions no survivors were recorded for any susceptible population treated with 560 g a.e. ha^−1^ of glyphosate, whereas to completely control the shifting populations, 720 g a.e. ha^−1^ was necessary. This indicates that slightly higher doses, while remaining abundantly within the label recommendations, may be useful to control and hopefully reduce the evolution of resistance to this herbicide.

Guidelines for future herbicide-resistant weed management globally should focus on avoiding a general use of reduced herbicide, especially glyphosate [36]. Successful integrated weed management strategies should aim at decreasing weed seed banks and reducing herbicide use. This involves adjusting the herbicide doses applied to achieve both a reduction in the number of treatments as well as an increase in the number of weeds controlled by the treatments. In this context, this research provides useful information to avoid or slow down the selection of glyphosate resistance in *Lolium* spp. by establishing a threshold for identifying future shifts of susceptibility. 

## 4. Materials and Methods 

### 4.1. Plant Material

Seeds of *Lolium* spp. were collected in agricultural and non-agricultural sites including field margins, organic farms (winter cereals), conventional farms (winter cereals, sunflower and perennial crops) and roadsides (Table 1). When available, details of historical herbicide use on the sampled fields were recorded. Sampling sites covered all major Italian agricultural areas and were chosen according to the absence or moderate application of glyphosate during the last decade. Areas where glyphosate-resistant *Lolium* spp. had been already reported were excluded [21]. Preference was given to regions where *Lolium* spp. are widespread and potentially cause severe economic losses. In each site, seeds were randomly collected from at least 30 plants spatially distributed in a sampling area of about 400 m^2^. Although morphological traits showed a high variability among and within populations, all of them were classified as *L. rigidum* or *L. multiflorum* or intermediates between the two species (LOLSS) (Table 1). The standard susceptible population S-204L, collected more than 15 years ago and reproduced in the greenhouse of the Institute for Sustainable Plant Protection (IPSP)- CNR (45°21′ N, 11°58′ E) was also tested. 

After ripening, seeds were kept in paper bags and then stored in a cool chamber at 4 °C until use. 

### 4.2. Dose-Response Experiments

#### 4.2.1. Spring Dose-Response Experiment

Twenty populations were included in the spring experiment (Table 1a), twelve putative susceptible populations, six known resistant and two “shifting” populations [19,31]. To break dormancy, seeds were vernalized at 4 °C in Petri dishes on wet filter paper, in darkness for three days. They were then placed in transparent plastic dishes on 0.6% (wt/V) agar medium and placed in a germination cabinet at the following conditions: temperature (day/night) 25/15 °C, 12 h photoperiod with neon tubes providing a Photosynthetic Photon Flux Density (PPFD) of 15–30 μmol m^−2^ s^−1^. Nine germinated seedlings at similar growth stage were transplanted into pots (15 × 15 × 20 cm) filled with a standard potting mix (60% silty loam soil, 15% sand, 15% perlite and 10% peat). To better mimic field conditions, pots were kept outside in a semi-controlled environment, and the soil water content was maintained at or near field capacity. Temperature ranged day/night from 18.8 °C to 7.9 °C. The experimental layout was a completely randomized design of three replicates per dose (a total of 27 plants per dose). Eleven doses (geometrically distributed) of glyphosate (MON 79351) 480 g a.e. L^−1^ were considered: 45, 90, 135, 180, 270, 360, 450, 540, 720, 1080 and 1440 g a.e. ha^−1^. An untreated control was included for each population. Herbicide was sprayed when plants reached the stage BBCH 21 using a precision bench sprayer according to the following conditions: spray volume 200 L ha^−1^, pressure 215 kPa, speed 0.75 m s^−1^ using TeeJet nozzles TP11001-VH. Plant survival and fresh weight were recorded four weeks after the treatment. 

#### 4.2.2. Autumn Dose-Response Experiment

Thirty populations sampled as described in Section 4.1 in different Italian agricultural environments (Figure 5) were included in the autumn experiment (Table 1, a in bold and b). All populations were putatively susceptible to glyphosate in order to calculate the threshold value of the sensitivity analysis. To compare the data of the two experiments, 12 susceptible populations (in bold in Table 1a) selected for the spring experiment were also included in the autumn experiment. Seeds preparation, seedlings transplanting and growth conditions, as well as treatment conditions, were as described in Section 4.2.1. Temperature ranged day/night from 20.3 °C to 11.2 °C. Plant survival and fresh weight were recorded four weeks after treatment.

### 4.3. Statistical Analyses

The mean survival and fresh weight per dose were expressed as a percentage of the untreated control. The ED_50_, GR_50_, ED_90_, GR_90_ and relative standard errors for the mean percentage of plant survival and fresh weight were calculated by non-linear regression analysis performed using the macro BIOASSAY^®^ developed by Onofri (2005) [37] and running in Windows Excel^®^. The macro is based on a log-logistic equation to fit the data: Y = C + {(D − C)/[1 + (x/I_50_)b]} where Y is the fresh weight or survival, C and D are the lower and upper asymptotes at higher and zero doses, respectively, I_50_ (or I_90_) is the herbicide dose resulting in a 50% (or 90%) reduction in plant biomass or survival, i.e., ED_50_ and GR_50_, respectively (or ED_90_ and GR_90_, respectively), b is the slope. The procedure estimates the standard error of the parameters and performs the Box-Cox power transformation family. For biological reasons and to improve the estimates of other parameters, the upper and lower asymptotes of survival data were forced to 100 and zero, respectively, whereas no parameters were constraints considering fresh weight data. Data of each population were first analyzed as a single curve to estimate the parameters and then all curves were regressed together. The data of the two experiments were analyzed separately. No parameters were fixed in the first analyses, and this complex model was then compared with progressively simplified models having common parameters among curves. The lack-of-fit F-test was performed at each step, and the simplification stopped when a significant lack of fit occurred. 

The Box and Whisker plot analysis was used to identify possible outliers and extreme values described as the values greater, or lower, than 1.5 and 3 times the value of the Box, respectively [38]. The analysis was repeated excluding the outlier values at each step until no outliers were detected.

In order to determine if the data of the two experiments could be compared, an ad hoc lack-of-fit F-test was applied to the data of each population included in both experiments by comparing data singularly. 

R software 3.2.5 and, in particular, the package *NbClust* [39] was applied to cluster ED_50_ and ED_90_ data. The package compares 30 different clustering methods and chooses as best the partition proposed by the majority of the methods.

### 4.4. Sensitivity Line Calculation

The mean values of EDs and GRs of the two experiments were compared using a paired t-test at *p* < 0.05 (excluding populations with extreme or outlier values in the Box and Whisker analyses).

The threshold value of the sensitivity analysis (sensitivity line) was calculated as the mean of the ED_90_ values with a slight modification compared to the method used by Paterson et al. [26] across biologically relevant populations and experiments. A population can be considered as shifting if the difference between the sensitivity line and population is greater than 2x and resistant if the difference between the sensitivity line and population is greater than 3x. To illustrate the variability of the response of populations to glyphosate, the ED_50/90_ variation among populations and the S.I. were calculated as the ratio between ED_50_ and ED_90_ of each population and the ED_90_ of the most tolerant and most sensitive populations, respectively. 

## Figures and Tables

**Figure 1 plants-09-00165-f001:**
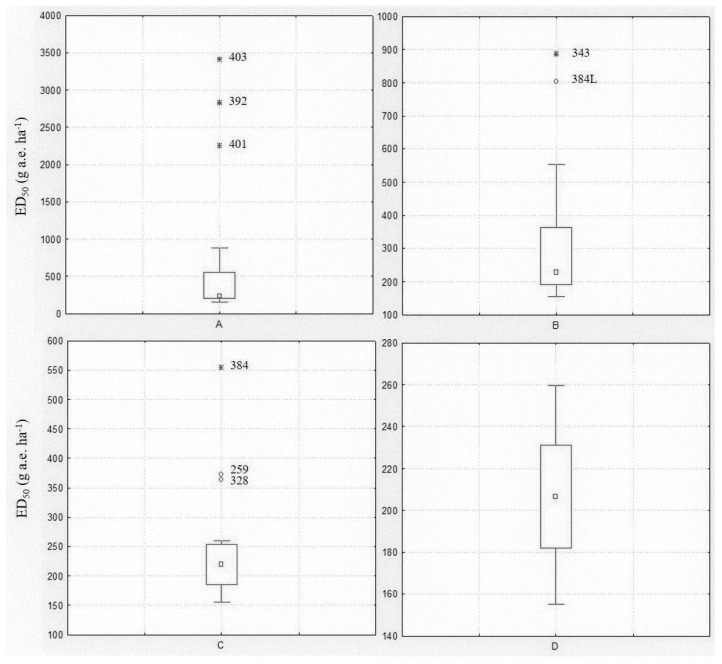
Box and Whisker plots illustrating the range of plant control (ED_50_) for the twenty *Lolium* spp. populations included in the spring experiment: (**A**) all populations were included; (**B**) extreme values of Box Plot A were excluded; (**C**) extreme and outlier values of Box Plot B were excluded; (**D**) extreme and outlier values of Box Plot C were excluded. The central point is the median, the box represents the 25–75 percentiles and bars the non-outlier range, ○ and * represent outliers and extreme values, respectively. Population codes excluded during the analysis are reported.

**Figure 2 plants-09-00165-f002:**
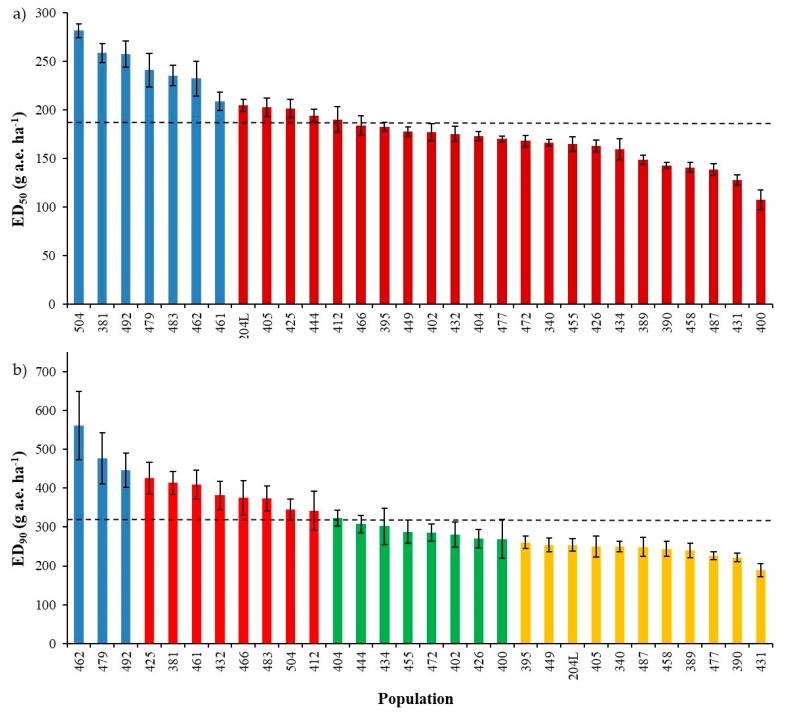
Response of thirty *Lolium* spp. populations included in the autumn experiment estimated by (**a**) the dose controlling 50% of plants (ED_50_) and (**b**) 90% of plants (ED_90_). Bars indicate standard errors (SE). Dashed horizontal lines represent the mean values of ED_50_ and ED_90_ in graphs (**a**) and (**b**), respectively. Different colours represent the subdivision of the populations obtained with the cluster analysis: (**a**) two clusters, (**b**) four clusters.

**Figure 3 plants-09-00165-f003:**
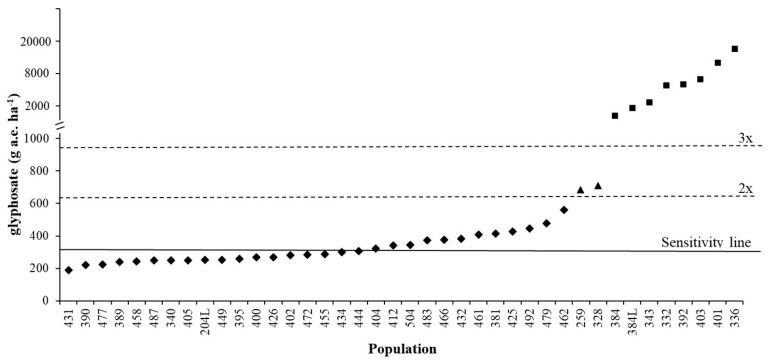
ED_90_ of different *Lolium* spp. populations: ♦ S populations tested in the autumn dose-response experiment, ▲ shifting populations tested in the spring dose-response experiment, ■ resistant populations tested in the spring dose-response experiment and/or discussed in [19] and [31]. Continuous line at 317 g a.e. ha^−1^ represents the sensitivity line calculated in this research, dashed lines represent 2x and 3x the sensitivity line value.

**Figure 4 plants-09-00165-f004:**
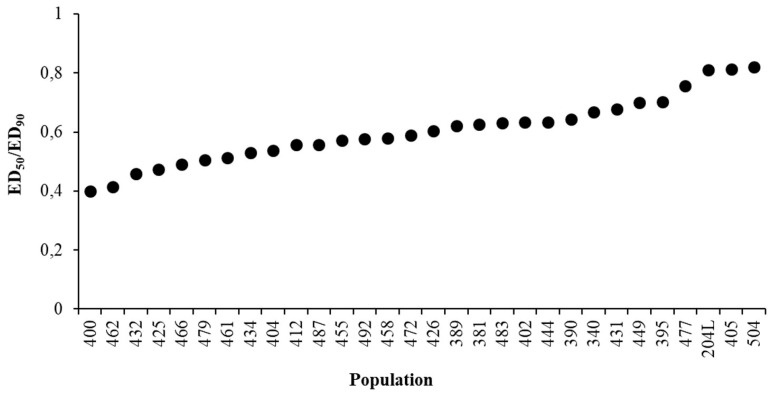
Variations of the ED_50_/ED_90_ ratio among populations.

**Figure 5 plants-09-00165-f005:**
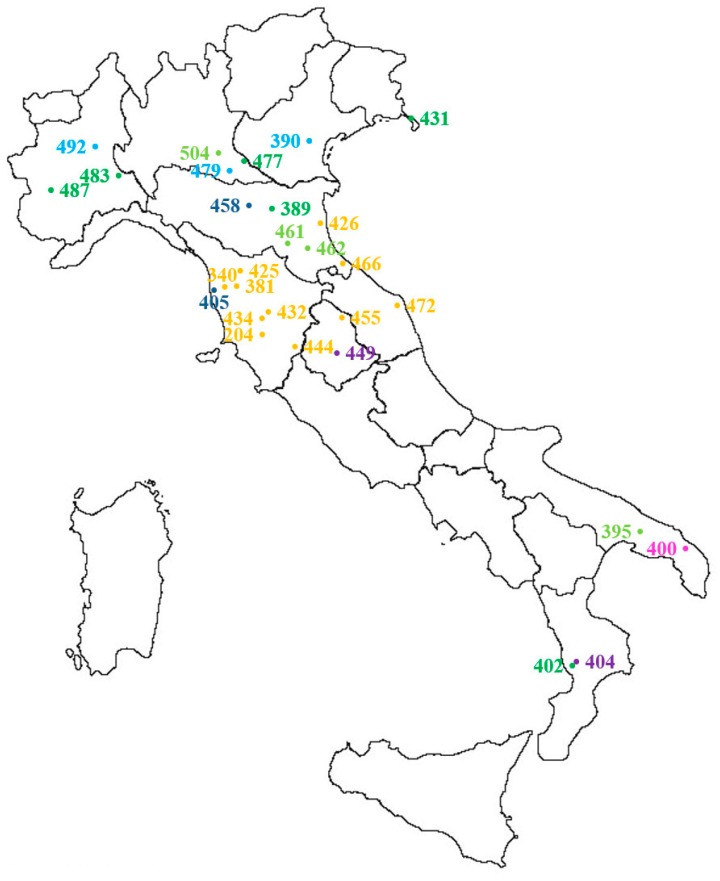
Distribution and origin of *Lolium* spp. populations included in the autumn dose-response experiment (see also Table 1b and populations in bold in Table 1a): yellow = wheat, dark blue = Lucerne, light blue = field margin, dark green = meadow, light green = roadside, purple = perennial, pink = set aside. For IPSP S check 204L, the origin of the original population is reported (Civitella Paganico, GR). Population 412 is not included because it comes from a commercial seed stock.

**Table 1 plants-09-00165-t001:** Details of the populations tested in the spring (a) and autumn (a in bold and b) dose-response experiments: species (LOL = *Lolium*, RI = *rigidum*, MU = *multiflorum*, SS = *multi-species*), sampling year, population code (progressive number, which, together with the sampling year, uniquely identifies a population), geographical origin and crop or collection site, where available. S = susceptible and R = resistant (i.e., plant survival >20% at the field dose); L = population reproduced in Legnaro greenhouse.

Species	Population Code	Origin(Municipality)	Crop or Collection Site	Notes
(a)
LOL	RI	07	328	Santo Stefano Belbo	vineyard	Partially R [19]
**LOL**	**RI**	**08**	**204L**	**Legnaro**	**wheat**	**S check used by IPSP-CNR**
LOL	SS	08	259	Cortona	wheat	Partially R [31]
**LOL**	**SS**	**08**	**340**	**Collesalvetti**	**wheat**	
LOL	SS	08	343	Pomarance	wheat	R pop. [31]
**LOL**	**SS**	**10**	**381**	**Pontedera**	**wheat**	
LOL	SS	10	384	Cascina	wheat	R pop. [31]
LOL	SS	11	384L	Cascina		Reproduced from glyphosate-resistant plants of pop. 10–384
**LOL**	**SS**	**10**	**389**	**Castenaso**	**meadow**	
**LOL**	**MU**	**11**	**390**	**Legnaro**	**field margin**	
LOL	SS	11	392	Palo del Colle	olive grove	R pop. (unpublished data)
**LOL**	**RI**	**11**	**395**	**Acquaviva delle Fonti**	**roadside**	
**LOL**	**RI**	**11**	**400**	**Torchiarolo**	**set aside**	
LOL	SS	11	401	Lamezia Terme	olive grove	R pop. (unpublished data)
**LOL**	**RI**	**11**	**402**	**Lamezia Terme**	**meadow**	
LOL	SS	11	403	Cascina	sunflower	R pop. [31]
**LOL**	**RI**	**11**	**404**	**Lamezia Terme**	**olive grove**	
**LOL**	**SS**	**11**	**405**	**Livorno**	**lucerne**	
**LOL**	**MU**	**11**	**412**		**Commercial turf seed**	
**LOL**	**SS**	**11**	**425**	**Cascina**	**wheat**	
(b)
LOL	MU	12	426	Ravenna	wheat	
LOL	SS	12	431	Duino Aurisina	meadow	
LOL	SS	12	432	Siena	wheat	
LOL	SS	12	434	Sovicille	wheat	
LOL	SS	12	444	San Casciano dei Bagni	wheat	
LOL	MU	12	449	Marsciano	vineyard	
LOL	SS	12	455	Gubbio	wheat	
LOL	MU	12	458	Montecchio Emilia	lucerne	
LOL	SS	12	461	Brisighella	roadside	
LOL	MU	12	462	Forlì	roadside	
LOL	MU	12	466	Coriano	wheat	
LOL	SS	12	472	Osimo	wheat	
LOL	SS	12	477	Pozzolengo	meadow	
LOL	SS	12	479	Cremona	field margin (maize)	
LOL	MU	12	483	Alessandria	meadow	
LOL	SS	12	487	Saluzzo	meadow	
LOL	SS	12	492	Cigliano	field margin	
LOL	SS	12	504	Pontoglio	roadside

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
