# Peer review of "Sensitivity Analysis of Italian Lolium spp. to Glyphosate in Agricultural Environments"

_plants, 2020, doi:10.3390/plants9020165_

Round 1
Reviewer 1 Report
The manuscript by Panozzo et al. seeks to develop an understanding of the background sensitivity of two ryegrass species to glyphosate herbicide, as tolerance / resistance to glyphosate is developing in local populations in Italy.
Clearly, a lot of work has went into this, and it has been carefully done. The question is relevant, although this is clearly not hypothesis-driven research, and little in the way of mechanisms to resistance are investigated or discussed. The paper is designed to be a self-contained study, and in that goal it is very successful. The English throughout is of a very good, although there are a couple of phrases which I felt needed a little more work.
However, the paper does have some weaknesses, mainly in its stucture.
Plants uses an Introduction - Results - Discussion - Materials and Methods structure, and in my opinion, writing for this structure needs to be quite careful. My biggest complaint about the manuscript is really in the transition from the Introduction to the Results. I felt like you need more explanatory text in this section to guide the reader through the logic of the paper.
Putting this in direct terms;
End of the Introduction
"The aim of this research was to determine the variation in Lolium spp. response to glyphosate in Italian populations collected in agricultural environments"
goes to the Results;
"The lack-of-fit F-test on both plant survival and fresh weight...."
There's no obvious connection between these two sentences. I strongly wondered, "what plants?" "what F-test?" "what treatments?" "what's being measured, and how?" "why are these things being measured?" "what are the authors trying to measure?"
I think you need a paragraph which starts
"An experiment was designed to test the effects of glyphosate concentration on plant survival and growth..."
This is quite consistent throughout the Results - I never really knew why you were doing things. I constantly felt lost. I strongly urge the authors to put in some more signposting for readers.
Minor points
Throughout the manuscript there is a lack of clear definitions on abbreviations used
g a.e. ha-1 is never defined (mainly the a.e. bit). This may confuse more casual readers.
ED comes up in the text (line 94) until the figure legend of Fig. 2. It should be defined in the text at first usage.
R populations (line 99) has the same problem. I assume you mean that the statistical populations which were defined by an R module, but since your Materials and Methods aren't until the end of the paper, you need to make this clearer. Possibly just by saying they are separate populations (i.e. leaving out "R"), or by clearly explaining what was done.
Or does it mean "resistant"? Table 1 fig legend defines R this way.
GR50 suffers the same lack of definition
The abbreviation WAT is only used twice (line 290 and 300), and is probably unnecessary.
Ln. 17; Lolium spp. is around --> Lolium spp. was around
Ln 21: control - weak word. Specifying what you mean by that would be better
Ln 21; BBCH21 - undefined abbreviation. OK for an abstract, but should at least be defined or referenced in the text.
Ln 29; "spread worldwide" --> has a global distribution
Ln57; "in the two" --> "in two"
Ln 59; Put a period / full stop after the reference [18]. Delete "i.e.", and capitalize In
Ln 60; "in the past the rate of" --> "In the past, a rate of"
Ln 61; Sentence beginning "When exposed to..." should read
"Exposure to recurrent selection at sub-lethal glyphosate doses can result in a shift towards resistance within a few generations."
Ln 63; I think a comma after the word "Lolium" would improve readability
Ln 68; I don't like the word "involved" here. Maybe "affected"? "have resistant populations"?
Ln 71; shifts of susceptibility --> shifts in susceptibility
Ln74; of target species --> of a target species / of the target species
Ln 74; perhaps "species" needs an apostrophe? "target species' susceptibility?"
Ln 75; comma after "ingredient"
Ln 76; "in a decision-making process" --> "in decision-making processes"?
Ln 77; "acti"? "active"?
Ln 77: "is not correct" --> "is difficult"? "is impractical"?
Ln 81; "literature dealt" --> literature has dealt
Ln 96; exclude the outliers --> exclude outliers
Ln 96; "the following experiment". What following experiment? The following data? I think I just don't know what you're trying to do, so I'm confused.
Ln 104; there is a closing set of inverted commas (i.e. shifting"), but I didn't see an opening set.
Ln 105; show --> reveal?
Fig 1 resolution / clarity is really bad in the pdf
Ln 135: the two experiment --> the two experiments
Ln 136; Data of none of the populations could be pooled --> The population data could not be pooled ...
Ln 146; whereas data --> whereas the data
Ln 148; "origin of the population" -- defined how? geographical origin?
Ln 160; proved to be not significantly different --> No significant differences were found between ....
Ln 166; species is around --> species was around
Ln212: agricultural realities --> agricultural systems?
Ln 215; varied with the seasons --> is known to vary seasonally
Ln 217; intermediates, neither --> intermediates, and neither
Ln 218; correlation is present --> correlation present (or "Most probably, if a correlation is present")
Ln 219: "needs to be assessed as observed" - I don't know what this means. Do you mean using a methodology similar to that used here?
Ln220; from "for example", sentence needs to be re-phrased
Ln223; survived plants --> surviving plants
Ln 231; Data variability increased with ED, which may indicate...
Ln233; delete "also" ("also considers also")
Ln 233; is --> was
Ln 234; what is "they"?
Ln 237; BBCH 21 - undefined
Ln 238; "has to be at least..." --> should be 700 g a.e. ha-1 of glyphosate or higher
Ln 293; Thirty populations sampled --> Thirty populations were sampled
Ln 292; do you mean section 2.1? or 4.2.1? (since this is 4.2.2, and section 2.1 is in the results, it is a little confusing)
I do think your paper is reasonably easy to publish with a little improved signposting. I would like to specifically point out that I enjoyed reading your Introduction (but not the Results!)
I think a better explanation of the experiments done and the rationale in the final paragraph of the introduction and throughout the results would greatly help the manuscript.
Reviewer 2 Report
Sensitivity analysis of Italian Lolium spp. To glyphosate in agricultural environments. by Panozzo et al.
At first, I should say that this paper based on steady works is very important. As described, determination of field dose of each herbicide for a specific target should be carefully done with well-designed monitoring. Long term usage of particular herbicide with insufficient (sub-lethal) concentration definitely causes problems. I agree. There is no doubt.
But I think this manuscript still needs intense editing by English native. There are many points to be amended. Because of these, I cannot evaluate this important work at this time. I try to raise some points to be revised, but those are not all.
Most importantly, many of readers of this journal are not necessarily herbicide-specialists. There are some points that had better be revised for help understanding of general plant scientists. I would be happy if some points I raised are helpful for authors to revise the MS. After publication of the manuscript, significant numbers of people will be influenced by the specific numbers of herbicide dose that are indicated in this work. The influence could be associated with significant responsibility. So, please take enough cares about data that authors are going to present.
Introduction
L29-30. Grammatically, collect?
L41. What HRAC stands for?
L55. “orchards” are not “crops”.
L77. acti >> active?
Results
L94. Even for plant researchers, some technical terms do not seem to be familiar. For instance, definitions for EDs and ED50 should be explained in the first time. Please clarify what the abbreviation, ED, stands for. The explanation also came after the first appearance in the text (in legend of Figure 2).
L112. GR as well.
Table 1.
What RI, SS, and MU mean? Population code is complicated, too. What “L” indicates?
Round 2
Reviewer 1 Report
The manuscript is much improved, and I found it much easier to read.
Minor points
Ln 107; “collected in Italian agricultural environments” in --> from
Ln 119; “as single curves” --> individual curves OR individually
Ln 122; remove “therefore”
Ln 124; until no outliers --> no further outliers
Ln 128; “The third analysis considered an ED50 range of 155-560 g a.e. ha-1 and highlighted three other outlier populations (384, 130 259 and 328) (Figure 1C), two included in the experiment as R checks and the two "shifting” populations (Table 1).” I don’t understand this. You found 3 outlier populations, but then list 2 (R checks) and 2 (shifting). Is it these 3?
Ln 132; “The Box & Whisker analysis, repeated also for the ED90 and GRs values (data not shown), confirmed that all eight populations included in the experiment as resistant or partially resistant to glyphosate, were confirmed as having a reduced sensitivity or resistance to glyphosate.” Maybe it’s the second “confirmed”, but I find it hard to understand the final clause in relation to the one immediately before it. Probably the comma after (data not shown) is unnecessary.
Ln 178; varies from --> varied from
Ln 181; Cluster analyses was assessed in order to determine if a correlation was present --> Cluster analysis was used to determine whether a correlation was present / whether correlations were present
Ln 202; presumably a paired t-test?
There were a couple of other bits which were poorly phrased, so I'd suggest a line-by-line read through by the authors to clear up any confusing bits.
Reviewer 2 Report
I believe this steady work has importance in agriculture. The revised version certainly improved compared to the previous one. Jargons are now well defined, and I believe this make the MS more understandable for potential readers.
Author Response
No more comments to add, just thanks the reviewer for his/her comments